# CONCEPT FORGETTING VIA LABEL ANNEALING

## ABSTRACT

The effectiveness of current machine learning models relies on their ability to grasp diverse concepts present in datasets. However, biased and noisy data can inadvertently cause these models to be biased toward certain concepts, undermining their ability to generalize and provide utility. Consequently, modifying a trained model to forget these concepts becomes imperative for their responsible deployment. We refer to this problem as *concept forgetting*. Our goal is to develop techniques for forgetting specific undesired concepts from a pre-trained classification model's prediction. To achieve this goal, we present an algorithm called **L**abel **AN**nealing (**LAN**). This iterative algorithm employs a two-stage method for each iteration. In the first stage, pseudo-labels are assigned to the samples by annealing or redistributing the original labels based on the current iteration's model predictions of all samples in the dataset. During the second stage, the model is fine-tuned on the dataset with pseudo-labels. We illustrate the effectiveness of the proposed algorithms across various models and datasets. Our method reduces *concept violation*, a metric that measures how much the model forgets specific concepts, by about 85.35% on the MNIST dataset, 73.25% on the CIFAR-10 dataset, and 69.46% on the CelebA dataset while maintaining high model accuracy. Our implementation can be found at this following link: https://anonymous.4open.science/r/LAN-141B/

## 1 INTRODUCTION

The superior performance capability of deep learning systems is primarily attributed to their ability to learn various concepts inherent in the dataset. For instance, advancements in face recognition systems (LeCun et al., 1998; Krizhevsky et al., 2009; He et al., 2016) can be largely attributed to their ability to discern and characterize different semantic features from facial images, such as age, gender, and facial hair characteristics, etc. Similarly, in tasks involving image and text generation (Ramesh et al., 2021; 2022; Rombach et al., 2022), the ability to learn varied concepts present in the dataset, enables the generation of realistic and diverse outputs. Consequently, the efficacy of these models relies upon the acquisition of learned concepts inherent within the dataset. Nevertheless, when the dataset is tainted with noisy samples or biased concepts (Tommasi et al., 2017), these models are susceptible to learning such undesired biased concepts. For example, suppose we are learning a model to predict whether a person should get a bank loan or not. Such a model should not depend on the gender or race of the person. However, it is possible that the machine learning model might inadvertently use these features to make predictions, which is highly undesirable. As a result, there emerges a pressing necessity to forget or unbias the undesired biased concept from these trained models to ensure their reliable and accountable deployment. Apart from removing biases, forgetting concepts can prove beneficial in topics such as domain generalization. For example, envision a CelebA (Liu et al., 2015) image classifier that heavily relies on *background color* as a distinguishing feature to classify different celebrities, limiting its ability to generalize effectively. Therefore, in such scenarios, rapidly forgetting only undesired concepts from a pre-trained model, without affecting the ability of the model to use other features, can improve the model's fair decision-making and generalization capabilities.

To make a pre-trained model forget a concept, we start by asking the following question - *what is meant by forgetting?* Our definition of forgetting is motivated by the fact that in the case of humans, if one forgets a concept, the forgotten concept doesn't affect one's decision-making. Thus *concept forgetting* within the context of machine learning entails ensuring that a model's predictions

become entirely independent of the targeted forgetting concept. However, achieving this task presents challenges, as the goal is to forget a specific undesired concept without adversely affecting the ability of the model to use other concepts. This challenge is underscored by the phenomenon of *catastrophic forgetting* observed in the literature (McCloskey & Cohen., 1989; Goodfellow et al., 2014; Kirkpatrick et al., 2017; Ginart et al., 2019) in similar contexts, where adapting a model for new tasks (in our case task of *concept forgetting*), can significantly degrade performance. Thus, to explore the extent of forgetting concepts in pre-trained models, we pose the following challenge:

*Can we efficiently modify a pre-trained model to forget (unbias) an undesired (biased) concept while maintaining its performance?*

Before we proceed further, we first state the differences between concept forgetting and machine unlearning, the latter of which has been recently used to remove the effect of certain training examples from the model.

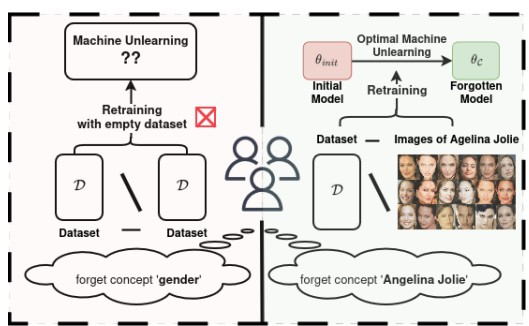

Figure 1: **Machine unlearning vs concept forgetting:** In the first scenario (on the left), such as gender, machine unlearning fails. This failure occurs because the dataset includes only males and females, making it impossible to retrain the model without the gender concept. In the second scenario (on the right), when a user requests the removal of concepts related to "Angelina Jolie," unlearning methods, like the optimal retraining approach, can be used successfully.

**Machine unlearning vs. concept forgetting:** *Machine unlearning* (Cao & Yang, 2015; Xu et al., 2020; Nguyen et al., 2022) aims to remove the influence of particular subsets of training data from a model so that the unlearned model mirrors the behavior of a retrained model that is trained from scratch without the undesired data subset. The best method to achieve this is by retraining the model from scratch without the unwanted data, although this process can be computationally expensive (Ginart et al., 2019; Sekhari et al., 2021). However the goal of *concept forgetting* is to make a model's predictions entirely independent of the targeted forgetting concept. The formal definition of *concept forgetting* is given in Sec. 3.1. Given this, we note that machine unlearning and concept forgetting as two different problem scenarios (see Figure 1). For example, consider the CelebA dataset which contains images of celebrities. Suppose we would like to make the model forget the concept of gender. A machine unlearning approach should remove the influence of all examples that have gender and

produce an unlearned model that is equivalent to retraining the model on the empty dataset as all CelebA dataset images have gender. However, as we show later, concept forgetting can be used to remove the bias caused by gender concept. We do note that machine unlearning can be potentially used to forget small concepts, which are only present in a subset of examples. For example, if we want to eliminate the concept of a particular celebrity, from a classifier trained on the CelebA dataset, we can retrain the model without the images of that celebrity. However, such applications are limited and our proposed algorithm works for removing or unbiasing the dependence of undesired concepts from the model's predictions.

**Our contributions:** Our contribution can be summarized as follows:

• We introduce the framework of *concept forgetting* from pre-trained classification models. Motivated by works in fairness (Dwork et al., 2012; Hardt et al., 2016; Lowy et al., 2021), we measure the bias or the dependence of a model on a concept, based on *concept violation*, which empirically quantifies the extent to which the model remains neutral towards a concept for its predictions.

• We propose an algorithm called **L**abel **AN**nealing (**LAN**). *LAN* employs an iterative approach, where each iteration redistributes the class labels of data points containing forgetting concepts to the most probable class labels, thus creating pseudo-labels. This realignment ensures that the distribution of pseudo-labels for each concept value matches the class distribution predicted by the current iteration's model. The method draws an analogy to the term *annealing* frequently employed in material science. It denotes the controlled redistribution of atoms within a solid material under specific temperature conditions to attain an equilibrium state, which inspired our method's nomenclature. This strategy

not only aids in mitigating the influence of the undesired concept on the model's predictions but also is computationally efficient. This method necessitates minimal epochs, sometimes as few as a single epoch, to diminish the reliance of the model's predictions on the forgetting concept, all the while maintaining the model's overall performance and generalization ability.

• We demonstrate the efficacy of our algorithms through detailed evaluations on various image classification datasets such as MNIST (LeCun et al., 1998), CIFAR-10 (Krizhevsky et al., 2009), miniImageNet (Vinyals et al., 2016), and CelebA (Liu et al., 2015) using state-of-the-art image classification models such as MobileNetV2, DenseNet-121, ResNet-50. Our method reduces (averaged over several concepts) concept violation, a metric that measures how much the model forgets specific concepts, by about 85.35% on the MNIST dataset, 73.25% on the CIFAR-10 dataset, 17.05% on the miniImageNet dataset, and 69.46% (averaged over 81.34% for binary concepts and 63.52% for multi-level concepts forgetting) on the CelebA dataset while maintaining high model accuracy.

## 2 RELATED WORKS

### 2.1 FAIRNESS

Fairness in machine learning systems is an important research area aimed at ensuring that system predictions are both accurate and fair across different groups (based on their features) of data points. Earlier works (Dwork et al., 2012) initially proposed the notion of *demographic parity* as a preliminary definition of fairness. According to this concept, a machine learning algorithm satisfies demographic parity if the predicted target is independent of sensitive attributes. However, promoting demographic parity may lead to diminished performance, particularly if the true outcome is not independent of sensitive attributes. To address this, subsequent works (Hardt et al., 2016) introduced a relaxed notion of fairness based on *equalized odds* and *equal opportunity* definition. Recent works (Kamishima et al., 2011; Feldman et al., 2015; Zafar et al., 2017; Donini et al., 2018; Mary et al., 2019; Cho et al., 2020a; Jiang et al., 2020; Rezaei et al., 2020; Lowy et al., 2021) have explored incorporating different fairness notions during the training process itself. These methods incorporate regularization-based techniques based on different statistical distances between the distribution of the model's prediction and sensitive attributes.

Drawing inspiration from fairness notions, especially *demographic parity* (Dwork et al., 2012; Lowy et al., 2021), we propose that forgetting a particular concept can also be interpreted as achieving independence between the model's prediction and the undesired feature we aim to forget. However, methods focusing on enforcing fairness with respect to the forgetting concept require a large number of epochs to converge and can be computationally inefficient. For instance, according to state-of-the-art FERMI algorithm (Lowy et al., 2021), achieving $||\nabla \ell(\theta, x, y)|| \leq \epsilon$ where $\ell$ is the loss function requires approximately $\mathcal{O}(\frac{1}{\epsilon^4})$ iterations. Empirically, the convergence of the FERMI algorithm varies depending on the dataset and application, typically ranging from as few as 50 to as many as 2000 epochs (Lowy et al., 2021, Appendix E). Given our specific objective of forgetting only certain concepts from a model's parameters without affecting others, we aim to devise a more computationally efficient approach for concept forgetting from pre-trained models.

### 2.2 MACHINE UNLEARNING FOR TRAINING EXAMPLE REMOVAL

*Machine unlearning*, as described in the literature (Xu et al., 2020; Nguyen et al., 2022; Cao & Yang, 2015; Ginart et al., 2019; Golatkar et al., 2020a;b; 2021; Neel et al., 2021; Nguyen et al., 2020; Guo et al., 2020; Graves et al., 2021; Sekhari et al., 2021), involves intentionally erasing the impact of particular subsets of training data from a pre-trained model, addressing user privacy concerns. Here, the objective is to craft a computationally efficient method that produces an unlearned model that mirrors the behavior of the model that is trained from scratch, on the training dataset devoid of the sensitive data points. Although retraining serves as the optimal benchmark method for unlearning, this method becomes computationally impractical for large models and iterative unlearning demands (Cao & Yang, 2015; Ginart et al., 2019; Sekhari et al., 2021). Consequently, to address user privacy concerns, more efficient data deletion methods (Cao & Yang, 2015; Ginart et al., 2019) were devised, leading to the emergence of *machine unlearning*. The *machine unlearning* methods are broadly categorized into two types: *exact unlearning* (Wu et al., 2020) and *approximate unlearning* (Neel et al., 2021; Sekhari et al., 2021). *Exact unlearning* aims to completely eliminate the influence of

unwanted data from the trained model, while *approximate unlearning* methods only partially mitigate data influence, resulting in parameter distributions closely resembling the retrained model with minor adjustments. More sophisticated methods (Guo et al., 2020; Graves et al., 2021) have suggested using influence functions, but these are computationally demanding and limited to small convex models. To extend unlearning techniques to non-convex models like deep neural networks, (Golatkar et al., 2020a;b) introduced a scrubbing mechanism centered on the Fisher Information matrix.

As we noted earlier, concept forgetting and machine unlearning have fundamental differences ( Figure 1 demonstrates machine unlearning cannot be applied for a general concept forgetting setup) in their objectives. *Machine unlearning* seeks to forget specific data points while emulating the behavior of a retrained model, whereas *concept forgetting* aims for the model's predictive performance to become independent of the forgotten concept.

## 3 PRELIMINARIES AND BACKGROUND

### 3.1 PROBLEM FORMULATION

Unless otherwise specified, we consider the problem of multi-class classification throughout the paper. Let $\mathcal{Y} \triangleq \{0, 1, 2, \ldots, k-1\}$ denote the set of $k$ labels. Let $z = (x, y)$ denote a data point where $x \in \mathbb{R}^d$ is the feature and $y \in \mathcal{Y}$ is the label. A dataset $\mathcal{D} \triangleq \{z_i\}_{i=1}^{|\mathcal{D}|}$ is a set of samples sampled from the underlying data distribution $P_{xy} : \mathbb{R}^d \times \mathcal{Y} \to [0, 1]$. Let a categorical concept $\mathcal{C} : \mathbb{R}^d \times \mathcal{Y} \to \{0, 1, 2, \ldots, m-1\}$ be a mapping from the sample to the set of all possible values the concept can take. For example, if the concept is binary such as beard, it can take two values $\{0, 1\}$ ($m = 2$), which denotes the absence and presence of the beard, respectively. Similarly, if the concept is non-binary such as facial hair type, it can take multiple values $\{0, 1, 2, 3\}$ ($m = 4$) which signifies no facial hair, mustache, beard, and goatee respectively. Let $h_\theta : \mathbb{R}^d \to \Delta^{|\mathcal{Y}|}$ denote a classifier parameterized by $\theta \in \mathbb{R}^p$ where $\Delta$ is the probability simplex. This classifier takes a feature $x \in \mathbb{R}^d$ and predicts a distribution over the label space. Let $\hat{h}(\theta, z)$ denote a post-processing step on the classifier (e.g. argmax) where a hard label is inferred based on the probability over the labels.

**Definition 1.** *(Concept neutral):* *We call a classifier with parameter $\theta$ concept neutral with respect to a concept $\mathcal{C}$, if for all output class $y \in \mathcal{Y}$ and all possible concept values $c \in \{0, 1, 2, \ldots, m-1\}$,*

$$P_{xy}(\hat{h}(\theta, z) = y | \mathcal{C}(z) = c) = P_{xy}(\hat{h}(\theta, z) = y). \tag{1}$$

**Definition 2.** *(Concept violation):* *For a classifier $\hat{h}$ with a parameter $\theta$, we measure the violation of concept neutrality in terms of the total variation distance as follows:*

$$V(\theta, \mathcal{C}, P) \triangleq \frac{1}{m} \sum_{c=0}^{m-1} d_{TV}\Big( P_{xy}(\hat{h}(\theta, z) = y), P_{xy}(h(\theta, z) = y \mid \mathcal{C}(z) = c) \Big)$$

$$= \frac{1}{2m} \sum_{c=0}^{m-1} \sum_{y=0}^{k-1} \Big| P_{xy}(\hat{h}(\theta, z) = y) - P_{xy}(\hat{h}(\theta, z) = y \mid \mathcal{C}(z) = c) \Big|. \tag{2}$$

Note that $V(\theta, \mathcal{C}, P) \in [0, 1]$ and if a model is concept neutral, then $V(\theta, \mathcal{C}, P) = 0$. As the underlying data distribution $P_{xy}$ is unknown, we have only access to the dataset $\mathcal{D}$ to empirically estimate concept violation $V(\theta, \mathcal{C}, P)$ as follows:

$$\hat{V}(\theta, \mathcal{C}, \mathcal{D}) \triangleq \frac{1}{m} \sum_{c=0}^{m-1} d_{TV}\Big( \hat{P}_{\mathcal{D}}(\hat{h}(\theta, z) = y), \hat{P}_{\mathcal{D}}(\hat{h}(\theta, z) = y \mid \mathcal{C}(z) = c) \Big)$$

$$= \frac{1}{2m} \sum_{c=0}^{m-1} \sum_{y=0}^{k-1} \Big| \hat{P}_{\mathcal{D}}(\hat{h}(\theta, z) = y) - \hat{P}_{\mathcal{D}}(\hat{h}(\theta, z) = y \mid \mathcal{C}(z) = c) \Big|, \tag{3}$$

where $\hat{P}_{\mathcal{D}}(\hat{h}(\theta, z) = y) = \frac{1}{|\mathcal{D}|} \sum_{z \in \mathcal{D}} \mathbb{1}(\hat{h}(\theta, z) = y)$, $\mathcal{D}_c = \{z \in D : C(z) = c\}$, and $\hat{P}_{\mathcal{D}}(\hat{h}(\theta, z) = y | \mathcal{C}(z) = c) = \frac{1}{|\mathcal{D}_c|} \sum_{z \in \mathcal{D}_c} \mathbb{1}(\hat{h}(\theta, z) = y)$. Now given a pre-trained model with parameter $\theta^*$ and a

concept $\mathcal{C}$, the goal of a concept forgetting algorithm is to find the forgotten parameter $\theta_\mathcal{C}$ such that the algorithm has the following properties:

• **Minimize empirical concept violation:** The *empirical concept violation* metric $\hat{V}(\theta_\mathcal{C}, \mathcal{C}, \mathcal{D})$ measures how much 'neutral' is the forgotten model for the given concept. For an ideal forgotten model, this metric will be zero indicating that the model has forgotten the concept. Hence minimizing concept violation is an important criterion and our goal is to ensure $\hat{V}(\theta_\mathcal{C}, \mathcal{C}, \mathcal{D}) \ll \hat{V}(\theta^*, \mathcal{C}, \mathcal{D})$.

• **Minimize accuracy loss:** Any forgotten model $\theta_\mathcal{C}$ should exclusively erase the specified concept without erasing others, thereby enabling the retained generalization capabilities to persist. Hence minimizing loss of the forgotten model's test accuracy is one of the important criteria. Let $\ell(\theta, z)$ denote the loss of the model with parameters $\theta$ for a sample $z = (x, y) \in \mathcal{D}$. Hence our goal is to ensure $\mathbb{E}_{z \sim P_{xy}}[\ell(\theta_\mathcal{C}, z)] \approx \mathbb{E}_{z \sim P_{xy}}[\ell(\theta^*, z)]$.

• **Small time complexity:** In dynamic environments, rapid model adaptation and updating are vital to remove biases or outdated information. Concept forgetting algorithms aim to selectively forget a few concepts from pre-trained models without affecting others. Efficiency is critical, as prolonged training may erase previously learned concepts.

# 4 METHODOLOGY

## 4.1 LABEL ANNEALING (LAN) ALGORITHM

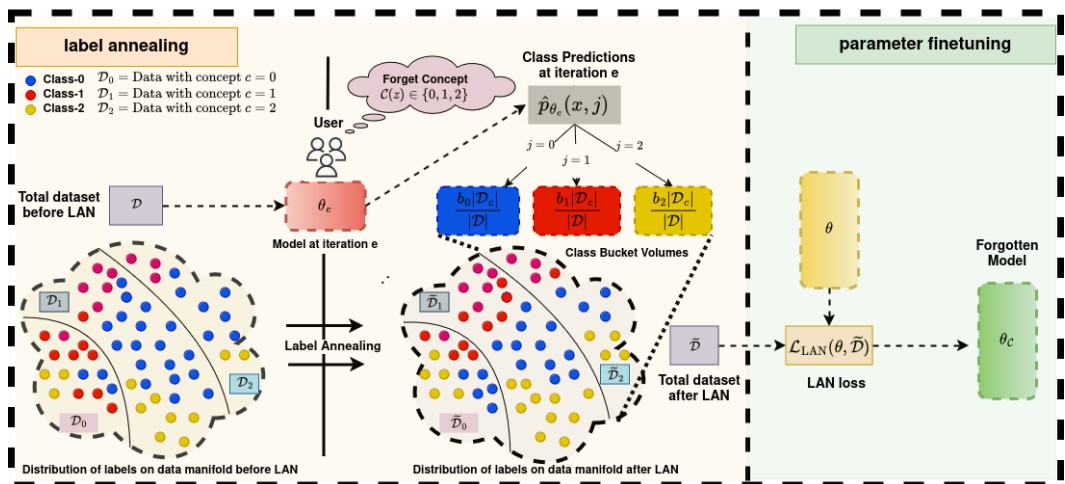

Figure 2: **Label Annealing (LAN) methodology** - The task involves forgetting the concept $\mathcal{C}(z) = c \in \{0, 1, 2\}$ from a classification task with data points labeled as $j \in \{\text{Class-0}, \text{Class-1}, \text{Class-2}\}$ (blue, red, and yellow, respectively). This iterative method runs $E$ iterations, where each $e^{th}$-iteration constitutes two stages: in first stage, known as *label annealing subroutine*, the labels within each concept data subset (e.g., $\mathcal{D}_0, \mathcal{D}_1, \mathcal{D}_2$) are redistributed based on the class prediction of $e^{th}$-iteration's model $\theta_e$, denoted as $\hat{p}_{\theta_e}(x, j)$, resulting in the label annealed dataset $\widetilde{\mathcal{D}}$. Subsequently at the next stage, termed as *parameter fine-tuning* using $\widetilde{\mathcal{D}}$, we minimize the loss function $\mathcal{L}_{\text{LAN}}(\theta, \widetilde{\mathcal{D}})$ to obtain final the concept forgotten model $\theta_\mathcal{C}$.

To achieve the forgotten model $\theta_\mathcal{C}$, we devise a method called Label Annealing (LAN). The overall methodology is shown in the Figure 2. At the heart of this algorithm is the *label annealing* subroutine, given in Algorithm 1. Given a model parameter $\theta_e$, training dataset $\mathcal{D}$, and particular concept $\mathcal{C}$ targeted for forgetting, this subroutine at a particular iteration $e$ creates a dataset with the same features as $x_i \in \mathcal{D}$ and with pseudo-labels $\tilde{y}_i$ such that the model $\theta_e$ has zero concept-violation on the newly created dataset $\widetilde{\mathcal{D}}$. Further to retain the model's overall performance, this assignment of pseudo-labels must result in a minimal change in empirical risk. Thus, we would like to change labels for the samples where changing the label does not significantly change the loss. To achieve these goals, the whole dataset is divided into concept data sub-groups $\mathcal{D}_c$ for each $c \in \{0, 1, ..., m - 1\}$.

---

**Algorithm 1** : **Label annealing subroutine**

**Input**: model parameter $\theta_e \in \mathbb{R}^p$, dataset $\mathcal{D}$, forgetting concept $\mathcal{C}$

1: For each class $j \in [k-1]$, let $b_j$ denote the number of samples $z \in \mathcal{D}$ with $\arg\max_j p_{\theta_e}(x, j) = j$.
2: **for** $c = 0, 1, \ldots, m-1$ **do**
3:     construct $\mathcal{D}_c = \{z \in \mathcal{D} : \mathcal{C}(z) = c\}, n_c = |\mathcal{D}_c|$
4:     For each sample $z_i \in \mathcal{D}_c$, the maximum probability assigned to sample $p_{\max}(x_i) = \max_j p_{\theta_e}(x_i, j)$. Sort $\mathcal{D}_c$ in decreasing order of $p_{\max}(x)$.
5:     Let $n_{c,j}$ denote the number of samples $z$ in $\mathcal{D}_c$ such that $\arg\max_j p_{\theta_e}(x, j) = j$.
6:     Let $\alpha_{c,j}$ be the number of samples in $\mathcal{D}_c$ assigned with class-j by the algorithm. Set $\alpha_{c,j} = 0 \forall j \in [k-1]$
7:     **for** $i = 1, 2, \ldots, |\mathcal{D}_c|$ **do**
8:         $\widetilde{y}_i \leftarrow \phi$
9:         $j^* \leftarrow \underset{j \in \{0, \ldots, k-1\}}{\arg\max} \, p_{\theta_e}(x_i, j)$
10:        **if** $\alpha_{c,j^*} < b_{j^*} \cdot n_c/|\mathcal{D}|$ **then**
11:            $\widetilde{y}_i \leftarrow j^*$
12:            $\alpha_{c,\widetilde{y}_i} \leftarrow \alpha_{c,\widetilde{y}_i} + 1$
13:        **end if**
14:    **end for**
15:    **for** $i = 1, 2, \ldots, |\mathcal{D}_c|$ **do**
16:        **if** $\widetilde{y}_i == \phi$ **then**
17:            $j^i \leftarrow \phi$
18:            **while** $\widetilde{y}_i == \phi$ **do**
19:                $j_i \leftarrow \underset{j \in \{0, \ldots, k-1\} \setminus j^i}{\arg\max} \, p_{\theta_e}(x_i, j)$
20:                **if** $\alpha_{c,j_i} < b_{j_i} \cdot n_c/|\mathcal{D}|$ **then**
21:                    $\widetilde{y}_i \leftarrow j_i; \quad \alpha_{c,\widetilde{y}_i} \leftarrow \alpha_{c,\widetilde{y}_i} + 1$
22:                **else**
23:                    $j^i \leftarrow j^i \bigcup \{j_i\}$
24:                **end if**
25:            **end while**
26:        **end if**
27:    **end for**
28: **end for**
29: **Output**: $\widetilde{\mathcal{D}} \leftarrow \{\widetilde{z}_i = (x_i, \widetilde{y}_i)\}_{i=1}^{|\mathcal{D}|}$

---

**Algorithm 2** : **Parameter fine-tuning**

**Input**: pre-trained parameter $\theta^* \in \mathbb{R}^p$, dataset $\mathcal{D}$, concept that needs to be forgotten $\mathcal{C}$, batch size $B$, learning rate $\eta$, number of iterations $E$, number of steps $T$.

1: **Initialize:** $\theta_e \leftarrow \theta^*$
2: **for** $e = 1, \ldots, E$ **do**
3:     $\widetilde{\mathcal{D}} \leftarrow \text{LAN}(\theta_e, \mathcal{D}, \mathcal{C})$
4:     **for** $t = 1, \ldots, T$ **do**
5:         Draw a random mini-batch of size $B$ from $\widetilde{\mathcal{D}}$ denoted as $\widetilde{\mathcal{D}}^b$
6:         $\theta_e \leftarrow \theta_e - \eta \nabla_\theta \mathcal{L}(\theta, \widetilde{\mathcal{D}}^b)$
7:     **end for**
8: **end for**
9: **Output**: $\theta_E$

---

Now for each of $\mathcal{D}_c$, the first term in Eq. 3 for a particular class label $j$ would be $\frac{b_j}{|D|}$ and second term would be $\frac{n_{cj}}{|n_c|}$ where $n_c = |\mathcal{D}_c|$, $b_j$ and $n_{cj}$ are the nos. of samples of class-$j$ predicted by the current model $\theta_e$ on dataset $\mathcal{D}$ and $\mathcal{D}_c$ respectively. In other words to make concept violation zero in $\mathcal{D}_c$, nos. of samples predicted class-$j$ in $\mathcal{D}_c$ i.e. $n_{cj}$ must be equal to $b_j \cdot \frac{n_c}{|D|}$. This is why we need to redistribute the labels of each class-$j$ in $\mathcal{D}_c$ without much affecting the model performance (empirical loss). Thus to achieve this dual objective of redistributing the labels without much affecting the empirical loss, we calculate $p_{\max}(x_i) = \max_j p_{\theta_e}(x_i, j)$ for each sample $z_i = (x_i, y_i) \in \mathcal{D}_c$, and then $\mathcal{D}_c$ is sorted in decreasing order of $p_{\max}(x)$. Thus in this sorted $\mathcal{D}_c$ (in the second for loop), each sample $x_i$ is initialized with label $\widetilde{y}_i = \phi$ and iteratively assigned the most probable label $\widetilde{y}_i = j^* = \arg\max_j p_{\theta_e}(x_i, j)$ until the no of samples in class-$j^*$ is less than $b_{j^*} \cdot \frac{n_c}{|D|}$. This second for loop ensures that reassigned labels don't change from the initial ones (this is why the deterministic assignment is done) on those data points where the model is confident (this is why $\mathcal{D}_c$ is sorted). Further in the subsequent steps (third for loop), the data points where the labels are unassigned i.e. $y_i = \phi$, it is assigned the subsequent (second or third and so on) most probable label class-$j$ if the no of samples in that assigned class-$j$ is less than $b_j \cdot \frac{n_c}{|D|}$. This loop tries to redistribute the labels where the model is not confident (low concept violation is achieved by trading off some accuracy). Subsequently, in the next stage of *parameter fine-tuning* (Algorithm 2), we fine-tune the $e^{th}$-model $\theta_e$ on the new dataset $\widetilde{\mathcal{D}} = \bigcup_{c=0}^{m-1} \widetilde{\mathcal{D}}_c$ to obtain the forgotten model $\theta_{e+1}$ by minimizing the Label Annealing loss function $\mathcal{L}_{\text{LAN}}(\theta, \widetilde{\mathcal{D}}) = \frac{1}{|\widetilde{\mathcal{D}}|} \sum_{c=0}^{m-1} \sum_{i=1}^{|\widetilde{\mathcal{D}}_c|} \ell(\theta, \widetilde{z}_i)$. We repeat this process for $E$ steps to get the final concept-neutral model $\theta_{\mathcal{C}}$. The value of E depends on the user's choice. However, to achieve concept forgetting with low computational complexity we experimented with E=1 (results of Table 1 and Table 2). Further ablation studies on $E = 2$ and $E = 4$ are given.

## 4.2 THEORETICAL ANALYSIS

In this section, we theoretically show that the proposed algorithm retains its accuracy if the original model has low concept violation. Recall that $\theta^*$ denotes the input to Algorithm 2 and $\theta_{\mathcal{C}}$ denotes the output of our algorithm. With this notation, we show the following result.

**Theorem 1.** *Let the loss function be bounded i.e., $\forall \theta, z \; \ell(\theta, z) \leq L$. If the fine-tuning reduces the loss on $\widetilde{\mathcal{D}}$ i.e., $\mathbb{E}\left[\mathcal{L}_{\widetilde{\mathcal{D}}}(\theta_{\mathcal{C}})\right] \leq \mathcal{L}_{\widetilde{\mathcal{D}}}(\theta^*)$, then*

$$\mathbb{E}\left[\mathcal{L}_{\mathcal{D}}(\theta_{\mathcal{C}})\right] \leq \mathcal{L}_{\mathcal{D}}(\theta^*) + 4 \cdot L \cdot E \cdot m \cdot \hat{V}(\theta^*, \mathcal{C}, \mathcal{D}), \tag{4}$$

*where the expectation is over the randomization in the stochastic gradients in Algorithm 2.*

The above bound implies that if the original concept violation is small, then the performance of the new model (trained on $\widetilde{\mathcal{D}}$) will not degrade significantly. In particular, if the original concept violation is zero, then the loss of the forgotten model is the same as the loss of the original model. Furthermore, while the upper bound degrades with $E$, as we show in experiments, the performance improves or remains the same with an increasing value of $E$. Due to space constraints, we provide the proof of the above theorem in Appendix A.

## 5 EXPERIMENTS AND RESULTS

### 5.1 DATASETS AND MODELS

For our experiments, we consider mainly forgetting two types of concepts: *binary-level concept* and *multi-level concept*. We have used different image classification models such as 2-layer-MLP (hidden layer size 500), Mobinetv2 (Sandler et al., 2018), Densenet-121 (Huang et al., 2017), Resnet-50 (He et al., 2016). Further to show the applicability of our method for different classification tasks across diverse datasets, we have used MNIST (LeCun et al., 1998), CIFAR-10 (Krizhevsky et al., 2009), miniImageNet (Vinyals et al., 2016), and CelebA (Liu et al., 2015) datasets. Different concept forgetting scenarios for $E = 1$ can be seen from Table 1 and Table 2. Further details about the datasets and models are included in the appendix section B.1.

### 5.2 EVALUATION METRICS

To evaluate the efficacy of any concept-forgetting algorithm we propose two different metrics as defined below:

• **Test empirical concept violation:** This metric denoted as $\hat{V}(\theta_{\mathcal{C}}, \mathcal{C}, \mathcal{D})$, is defined in equation 5, quantifies the concept neutrality of the forgotten model $\theta_{\mathcal{C}}$. Observe that $\hat{V}(\theta_{\mathcal{C}}, \mathcal{C}, \mathcal{D}) \in [0, 1]$, and a smaller $\hat{V}(\theta_{\mathcal{C}}, \mathcal{C}, \mathcal{D})$ signifies that the model is conceptually neutral regarding the forgetting concept $\mathcal{C}$. In the rest of the section, we denote $\hat{V}(\theta_{\mathcal{C}}, \mathcal{C}, \mathcal{D})$ as $\hat{V}_{\mathcal{C}}$.

• **Test accuracy:** This metric evaluates the generalization performance of the forgotten model, denoted as $A_{\mathcal{D}}$. Any concept forgetting algorithm mustn't render the initial model ineffective during the forgetting process. Therefore, maintaining accuracy close to that of the initial model $\theta^*$ is desirable.

### 5.3 BASELINES

According to our knowledge, this is the first work that introduces *concept forgetting* as a property of the forgotten model to induce independence from the forgetting feature during its prediction task. Thus for proper evaluation of our method, we adopt several baselines from fairness because these baseline methods also advocate for the independence of prediction and sensitive concept features. Here we have used particularly three baseline methods: (a) FERMI (Lowy et al., 2021) (b) Continuous-Fairness (Mary et al., 2019) and (c) Fairness-KDE (Cho et al., 2020b). We have used official implementation for both FERMI and Continuous-Fairness baselines while for Fairness-KDE an open-source implementation has been used. Further details about the baselines can be found in the appendix section B.3.2.

## 5.4 BINARY-LEVEL CONCEPT FORGETTING

We evaluated our approach for different classification scenarios to forget binary concepts with $c \in \{0, 1\}$ ($m = 2$). For example, as illustrated in Table 1, in the context of the MNIST digit classification problem, the objective is to forget a particular class digit concept e.g. class-3 data. Thus here $c = 0$ represents concepts of non-digit-3 data and $c = 1$ represents concepts of digit-3 data. Similarly, in the CelebA dataset for gender concept $c = 0$ represents male and $c = 1$ represents female. Table 1 shows the efficacy of our method for different concept-forgetting scenarios. In this case, the average reduction of concept violation is about 85.35% on the MNIST dataset, 73.25% on the CIFAR-10 dataset, 17.05% on the miniImageNet dataset, and 81.34% on the CelebA dataset, while retention of high model accuracy.

Table 1: Empirical concept violation $\hat{V}_C(\downarrow)$ and test accuracy $A_D(\uparrow)$ of the initial model and forgotten model via *LAN*. For the forgotten model, $\hat{V}_C$ reduced without significantly reducing $A_D$

| Dataset | Models | Task | Concept | Initial Model | | LAN | |
|---|---|---|---|---|---|---|---|
| | | | | $\hat{V}_C$ | $A_D$ | $\hat{V}_C$ | $A_D$ |
| CelebA | Resnet-50 | Young or not | Gender | 0.117 | 0.898 | 0.015 | 0.847 |
| | | Attractive or not | Gender | 0.2219 | 0.827 | 0.006 | 0.767 |
| | | Heavy makeup or not | Gender | 0.314 | 0.919 | 0.127 | 0.764 |
| miniImageNet | Resnet-50 | class 0-99 classification | Triceratops | 0.4991 | 0.9791 | 0.406 | 0.951 |
| | | | Bugs | 0.4966 | 0.9791 | 0.364 | 0.936 |
| | | | Fences | 0.4948 | 0.9791 | 0.466 | 0.96 |
| CIFAR-10 | Mobinet-v2 | class 0-9 classification | Bird | 0.440 | 0.928 | 0.103 | 0.871 |
| | | | Frog | 0.473 | 0.921 | 0.108 | 0.855 |
| | | | Truck | 0.472 | 0.921 | 0.113 | 0.855 |
| | Densenet-121 | class 0-9 classification | Bird | 0.445 | 0.923 | 0.152 | 0.869 |
| | | | Frog | 0.473 | 0.917 | 0.116 | 0.878 |
| | | | Truck | 0.472 | 0.917 | 0.147 | 0.861 |
| MNIST | 2-layer MLP | digit 0-9 classification | digit-3 | 0.479 | 0.974 | 0.055 | 0.883 |
| | | | digit-5 | 0.491 | 0.971 | 0.104 | 0.901 |
| | | | digit-8 | 0.470 | 0.976 | 0.081 | 0.889 |
| | Resnet-50 | digit 0-9 classification | digit-3 | 0.498 | 0.990 | 0.047 | 0.893 |
| | | | digit-5 | 0.492 | 0.992 | 0.078 | 0.905 |
| | | | digit-8 | 0.496 | 0.991 | 0.063 | 0.897 |

As there exists a trade-off between the two metrics of interest, for proper evaluation of our method with FERMI (Lowy et al., 2021), Continuous-Fairness (Mary et al., 2019), and Fairness-KDE (Cho et al., 2020b) baselines, concept-violation vs. accuracy trade-off plots are depicted in Figure 3. From these plots, it can be seen that for a particular accuracy, our method achieves lower concept violation (LAN trade-off curve lies below the others) than other baseline methods achieving better trade-off.

## 5.5 MULTI-LEVEL CONCEPT FORGETTING

Here the concept mapping function $C(.)$ denotes the multi-level concept with $c \in \{0, 1, \ldots, m-1\}$ ($m > 2$). Table 2 illustrates the performance of *LAN*—in reducing concept violation and maintaining

Table 2: Empirical concept violation $\hat{V}_C(\downarrow)$ and test accuracy $A_D(\uparrow)$ of the initial model and forgotten model via *LAN*. For the forgotten model, $\hat{V}_C$ reduced without significantly reducing $A_D$

| Tasks | Concepts | Initial Model | | LAN | |
|---|---|---|---|---|---|
| | | $\hat{V}_C$ | $A_D$ | $\hat{V}_C$ | $A_D$ |
| Young vs. Not-Young | Hair Color | 0.2 | 0.898 | 0.063 | 0.8626 |
| | Facial Hair | 0.11 | 0.897 | 0.0329 | 0.8921 |
| Attractive vs. Not-attractive | Hair Color | 0.195 | 0.827 | 0.083 | 0.7955 |
| | Facial Hair | 0.1716 | 0.827 | 0.076 | 0.8088 |
| Heavy Makeup vs. Not-Heavy Makeup | Hair Color | 0.157 | 0.92 | 0.073 | 0.881 |
| | Facial Hair | 0.316 | 0.919 | 0.077 | 0.844 |

test accuracy across various settings of concept forgetting from a pre-trained Resnet-50 model trained on the CelebA (Liu et al., 2015) dataset. In this context, concept forgetting involves the removal of certain features from a pre-trained classifier in the process of classifying other features. For example, while classifying samples as *young* vs. *not-young*, we aim to forget subtle feature concepts such as *hair color*, and *facial hair* from the pre-trained models. In this setting, the LAN algorithm reduces concept violation by about 63.52% without significantly affecting test accuracy. It can be seen from Figure 3 (a) and (b) that our method performs significantly better than other baseline methods.

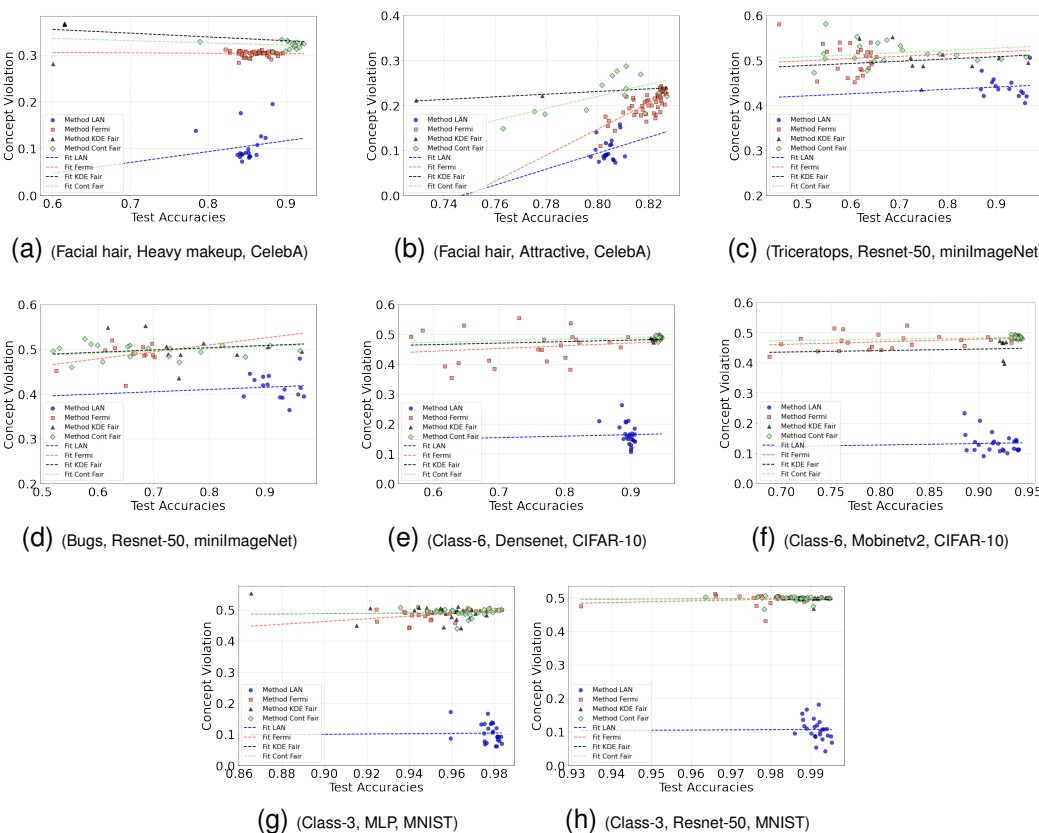

Figure 3: **Concept violation vs. accuracy trade-off:** We have plotted concept violation on the y-axis and accuracy on the x-axis. Each point represents an algorithm with a hyper-parameter, and the *Fit* line for an algorithm is obtained by a linear fit of all experiments corresponding to the algorithm. Figures (a) and (b) show forgetting facial hair removal from the task of heavy makeup and attractiveness classification on CelebA. Figures (c) and (d) show different concepts forgotten from pre-trained Resnet-50 on the miniImageNet dataset. Figures (e) and (f) show concept forgetting from pre-trained Densenet-121 and Mobinetv2, respectively, on the CIFAR-10 dataset. Figures (g) and (h) show class-3 concept forgetting from pre-trained MLP and Resnet-50 models, respectively, on the MNIST dataset. It can be seen that increasing accuracy increases concept violation. Thus, for a particular achievable accuracy, *LAN* achieves lower concept violation than other baseline methods.

## 5.6 ABLATION STUDY

Table 3 demonstrates the performance of the LAN method for different learning rates. As it can be seen as the learning rate increases the accuracy decreases while the concept violation decreases and then starts increasing again. Thus at higher accuracy regions, concept violation decreases along with accuracy whereas at lower accuracy regions concept violation increases with a decrease in accuracy. This suggests an optimal point lies in the trade-off curve where concept violation is low with a slight reduction of accuracy. Further in Figure 4, we demonstrate the effectiveness of *LAN* over multiple iterations (E=2, E=4). We present concept violation vs. accuracy trade-off plot to forget the facial

Table 3: Empirical concept violation $\hat{V}_C$ and accuracy $A_D$ for different learning rates.

| Dataset | Models | Concepts | Learning Rates | $\hat{V}$ | $A_D$ |
|---------|--------|----------|----------------|-----------|-------|
| MNIST | 2-layer MLP | Class-3 | 1.00e-07 | 0.476 | 0.973 |
| | | | 1.00e-05 | 0.091 | 0.884 |
| | | | 0.0001 | 0.055 | 0.883 |
| | | | 0.001 | 0.148 | 0.876 |
| | | | 0.005 | 0.257 | 0.842 |
| CIFAR-10 | Mobinetv2 | Class-6 | 1.00e-07 | 0.481 | 0.9253 |
| | | | 1.00e-05 | 0.141 | 0.861 |
| | | | 0.0001 | 0.108 | 0.856 |
| | | | 0.001 | 0.170 | 0.810 |
| | | | 0.005 | 0.210 | 0.6371 |
| miniImageNet | Resnet-50 | Class-3 | 1.00e-07 | 0.506 | 0.968 |
| | | | 1.00e-05 | 0.419 | 0.959 |
| | | | 0.0001 | 0.4166 | 0.8564 |
| | | | 0.001 | 0.453 | 0.477 |
| | | | 0.005 | 0.833 | 0.034 |
| CelebA | Resnet-50 | Facial Hair (Attractive) | 1.00e-08 | 0.234 | 0.826 |
| | | | 1.00e-06 | 0.103 | 0.817 |
| | | | 0.0001 | 0.076 | 0.800 |
| | | | 0.001 | 0.120 | 0.802 |
| | | | 0.01 | 0.320 | 0.680 |

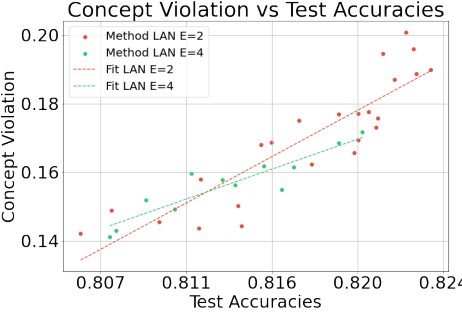

Figure 4: Multiple iteration concept violations vs. accuracy plots for *LAN* method

hair concept while classifying attractive vs. not attractive on the CelebA dataset. As $E$ increases at higher accuracy regions, the concept violation further decreases for the same accuracy value making the trade-off plot flatter.

## 6 CONCLUSION

In the pursuit of safer and more responsible machine learning, the elimination of undesired concepts from models is crucial. Our work focuses on efficiently removing these undesired concepts from pre-trained classification models, a task that is challenging due to the risk of *catastrophic forgetting*, which can render the model ineffective. To address this, we propose a computationally efficient algorithm termed *LAN* (Label Annealing) to create a forgotten model while preserving its ability to generalize. We define concept forgetting as the property of a model to disregard undesired concepts during its decision-making process and introduce *concept neutrality* as a necessary attribute of a forgotten model. To quantify the extent of *concept neutrality* in any model, we propose a novel metric called *concept violation*. Our experimental results demonstrate that our method effectively reduces *concept violation* while maintaining the model's performance across multiple concept-forgetting settings, various models, and datasets. Additionally, we acknowledge that our definition and method apply only to concept forgetting in classification models. Further research is needed to develop definitions and methods for concept forgetting that generalize to generative models as well.

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

## A  ANALYSIS

We recall some of the notation used in the algorithm. $b_j$ denotes the number of samples of class-$j$ in $\mathcal{D}$ predicted by the initial model. Thus, $n = |\mathcal{D}| = \sum_{j=0}^{k-1} b_j$. Similarly, let $n_{cj}$ for the number of samples of class-$j$ in $\mathcal{D}_c$. Hence, $n_c = |\mathcal{D}_c| = \sum_{j=0}^{k-1} n_{cj}$. Let the number of labels changed by Algorithm 1 (denoted by $\mathcal{A}$) in the total dataset $\mathcal{D}$ be $cl(\mathcal{A})$ and in concept data subset $\mathcal{D}_c$ be $cl(\mathcal{A})_c$. We first prove the following lemma.

**Lemma 1.** *Let $E = 1$. For any concept call $c$, the number of labels changed by Algorithm 1 is upper bounded by $cl(\mathcal{A}) \leq 2nm\hat{V}(\theta^*, \mathcal{C}, \mathcal{D})$.*

*Proof.* For a particular concept value $\mathcal{C} = c$ the label annealing subroutine Algorithm 1 changes the concept data subset to $\widetilde{\mathcal{D}}_c$ by redistributing the labels of the samples in $\mathcal{D}_c$. Let $T_{cj} = \min(n_{cj}, b_j \frac{n_c}{|\mathcal{D}|})$ and $\alpha_{cj}$ be the number of samples for class-$j$ in $\mathcal{D}_c$ assigned by the algorithm in current run. By closely observing algorithm 1 it can be said that the first phase (second *for* loop) algorithm tries to retain the original labels of the data until $\alpha_{c,j} < b_j \frac{n_c}{|\mathcal{D}|}$ while in the second phase (third *for* loop) the labels are assigned to other most likelihood classes. Thus the following propositions holds:

- If $T_{cj} = n_{cj}$, then the number of labels changed for class-$j$ in $\mathcal{D}_c$ termed as $cl(\mathcal{A})_{cj} = 0$.

- If $T_{cj} = b_j \frac{n_c}{|\mathcal{D}|}$, then the number of labels changed for class-$j$ is in $\mathcal{D}_c$ termed as $cl(\mathcal{A})_{cj} = \left| n_{cj} - b_j \frac{n_c}{|\mathcal{D}|} \right|$.

Hence, in the worst-case scenario number of labels changed for class-$j$ in $\mathcal{D}_c$ , $cl(\mathcal{A})_{cj} = \left| n_{cj} - b_j \frac{n_c}{|\mathcal{D}|} \right|$. Therefore,

$$cl(\mathcal{A})_c \leq \sum_{j=0}^{k-1} cl(\mathcal{A})_{cj} = \sum_{j=0}^{k-1} \left| n_{cj} - b_j \frac{n_c}{|\mathcal{D}|} \right|. \tag{5}$$

Now, for a particular concept $\mathcal{C} = c$ the empirical concept violation of the forgotten model $\theta_\mathcal{C}$ on $\widetilde{\mathcal{D}}_c$ is as follows:

$$\hat{V}(\theta^*, \mathcal{C} = c, D) = \frac{1}{2} \sum_{j=0}^{k-1} \left| \hat{P}_\mathcal{D}(\hat{h}(\theta^*, z) = j) - \hat{P}_\mathcal{D}(\hat{h}(\theta^*, z) = j \mid \mathcal{C} = c) \right| \tag{6}$$

$$= \frac{1}{2} \sum_{j=0}^{k-1} \left| \frac{b_j(\theta^*)}{|\mathcal{D}|} - \frac{n_{cj}(\theta^*)}{n_c} \right| \tag{7}$$

$$\geq \frac{1}{2n_c} cl(\mathcal{A})_c. \tag{8}$$

Hence, $cl(\mathcal{A})_c \leq 2n_c \hat{V}(\theta^*, \mathcal{C} = c, D) \leq 2n\hat{V}(\theta^*, \mathcal{C} = c, D)$. Summing over all concepts $c$ results in the lemma. $\qquad\square$

We will use the above lemma to prove our main result.

*Proof of Theorem 1.* We provide the proof for $E = 1$. The proof for larger values of $E$ follows by a telescoping sum of the epochs. Let's denote $\mathcal{L}_\mathcal{D}(\theta^*)$ and $\mathcal{L}_\mathcal{D}(\theta_\mathcal{C})$ denote the empirical losses of the pre-trained model and forgotten model on the initial dataset $\mathcal{D}$ respectively. Now following the notations from the above proof of Lemma 1 the number of labels changed in the whole dataset $\widetilde{\mathcal{D}} = \bigcup_{c=0}^{m-1} \widetilde{\mathcal{D}}_c$ is $cl(\mathcal{A})$. We now upper bound the empirical loss of $\theta_\mathcal{C}$ on $\mathcal{D}$ as follows:

$$\mathbb{E}\left[\mathcal{L}_{\mathcal{D}}(\theta_{\mathcal{C}})\right] = \mathbb{E}\left[\mathcal{L}_{\widetilde{\mathcal{D}}}(\theta_{\mathcal{C}}) + \mathcal{L}_{\mathcal{D}}(\theta_{\mathcal{C}}) - \mathcal{L}_{\widetilde{\mathcal{D}}}(\theta_{\mathcal{C}})\right] \tag{9}$$

$$= \mathbb{E}\left[\mathcal{L}_{\widetilde{\mathcal{D}}}(\theta_{\mathcal{C}}) + \frac{1}{n}\left[\sum_{z_i \in \mathcal{D}} \ell(\theta_{\mathcal{C}}, z_i) - \sum_{z_i \in \widetilde{\mathcal{D}}} \ell(\theta_{\mathcal{C}}, z_i)\right]\right] \tag{10}$$

$$\overset{(c)}{\leq} \mathbb{E}\left[\mathcal{L}_{\widetilde{\mathcal{D}}}(\theta_{\mathcal{C}})\right] + \frac{L}{n}\mathrm{cl}(\mathcal{A}) \tag{11}$$

$$\overset{(d)}{\leq} \mathcal{L}_{\widetilde{\mathcal{D}}}(\theta^*) + \frac{L}{n}\mathrm{cl}(\mathcal{A}) \tag{12}$$

$$= \mathcal{L}_{\mathcal{D}}(\theta^*) + \mathcal{L}_{\widetilde{\mathcal{D}}}(\theta^*) - \mathcal{L}_{\mathcal{D}}(\theta^*) + \frac{L}{n}\mathrm{cl}(\mathcal{A}) \tag{13}$$

$$\overset{(e)}{\leq} \mathcal{L}_{\mathcal{D}}(\theta^*) + \frac{2L}{n}\mathrm{cl}(\mathcal{A}) \tag{14}$$

$$\overset{(f)}{\leq} \mathcal{L}_{\mathcal{D}}(\theta^*) + 4Lm\hat{V}(\theta^*, \mathcal{C}, \mathcal{D}) \tag{15}$$

Here $(c)$ and $(e)$ holds as $\forall \theta$ if $z = \widetilde{z}$ then $\ell(\theta, z) = \ell(\theta, \widetilde{z})$ and the fact that $\ell(\theta, z) \leq L$. $(d)$ holds because of the assumption. Finally applying Lemma 1, we get $(f)$. □

# B ADDITIONAL DETAILS ABOUT EXPERIMENTS

## B.1 DATASETS AND MODELS

Here we have used four datasets as follows:

- **MNIST (LeCun et al., 1998):** The MNIST dataset consist of $28 \times 28$ gray-scale representing handwritten digits from 0 to 9. The MNIST dataset contains 6,000 images per digit class totaling 60,000 training samples and 1,000 images per digit class totalling 10,000 testing images.
- **CIFAR-10 (Krizhevsky et al., 2009):** The CIFAR-10 dataset consists of 60000 32x32 color images in 10 classes: airplane, automobile, bird, cat, deer, dog, frog, horse, ship, truck with 6000 images per class. There are 50000 training images and 10000 test images.
- **CelebA (Liu et al., 2015):** The Celeb Faces Attributes Dataset (CelebA) is a large-scale facial attributes dataset comprising over 200,000 celebrity images, each annotated with 40 attributes. This dataset features significant pose variations and background clutter. CelebA offers extensive diversity, a substantial quantity of images, and rich annotations.
- **miniImageNet (Vinyals et al., 2016):** Here we have used a smaller subset of the ImageNet dataset consisting of 50,000 training images and 10,000 testing images, evenly distributed across 100 classes. Here we have used an image dimension of $224 \times 224$ same as the original ImageNet data dimension.

**Models:** Further to evaluate our method to different models we experimented with a variety of models with different learnable parameter sizes such as 2-layer-MLP, Mobinet-v2 (Sandler et al., 2018), Densenet-121 (Huang et al., 2017), Resnet-50 (He et al., 2016). The 2-layer-MLP net has two hidden layers both having the size of 500. For Mobinet-v2, Densenet-121, and Resnet-50 we have taken Pytorch default models with pre-trained weights. For all of these models, the last layer is changed to an appropriate size suitable for the classification tasks.

## B.2 INITIAL TRAINING:

### B.2.1 INITIAL TRAINING ON MNIST

Here we have used 2-layer-MLP and Resnet-50 models for the classification tasks. For optimization, we have used the Adam optimizer with a learning rate of 0.001 on mean cross-entropy loss. All the models are trained for 5 epochs with a batch size of 64. The loss and accuracy curves can be seen in Figure 5.

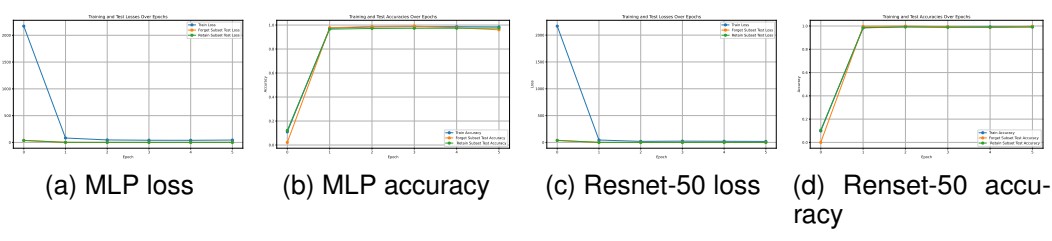

(a) MLP loss  (b) MLP accuracy  (c) Resnet-50 loss  (d) Renset-50 accuracy

Figure 5: Results of training the initial models on MNIST dataset

### B.2.2 INITIAL TRAINING ON CIFAR-10

Here we have used Mobinet-v2 and Densenet-121 models for the classification tasks. For optimization, we have used the Adam optimizer with a learning rate of 0.001 on mean cross-entropy loss. Mobinet-v2 and Densenet-121 models are trained for 60 and 20 epochs respectively with a batch size of 64. We have an early-stopping of 3 epochs for all the models. The loss and accuracy curves can be seen in Figure 6.

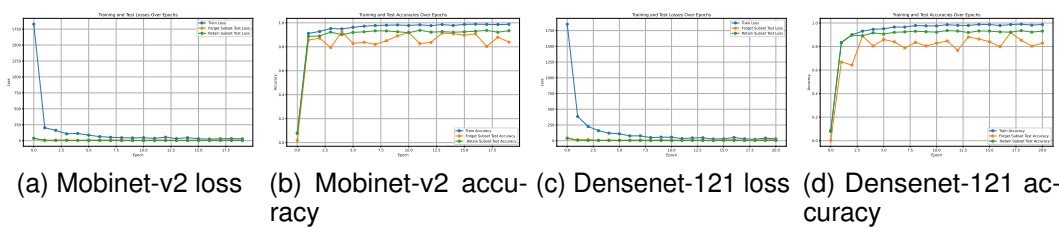

(a) Mobinet-v2 loss (b) Mobinet-v2 accuracy (c) Densenet-121 loss (d) Densenet-121 accuracy

Figure 6: Results of Training of the Initial Models on CIFAR-10 dataset

### B.2.3 INITIAL TRAINING ON CELEBA

Here we have used the Resnet-50 model for the classification tasks. As there are 40 attributes for classification we have trained 40 MLP heads for this. For optimization, we have used the SGD optimizer with a learning rate of 0.01 a learning rate scheduler with a decay of 0.1 every 30 steps, momentum of 0.9, and weight decay 1e-4 on total cross-entropy loss. Here Resnet-50 is trained for 90 epochs with a batch size of 256.

### B.3 TRAINING FOR CONCEPT FORGETTING

### B.3.1 LAN TRAINING

Our official codebase for LAN is available at the following link:`https://anonymous.4open.science/r/LAN-141B/`. Here we have used the label annealing methodology to finetune the pre-trained model for 1 epoch. We evaluated our method with both retraining and FERMI methodology in different forgetting settings. For our results the optimal hyper-parameters For different settings of forgetting we give the optimal hyper-parameters for our optimal results in the following tables

### B.3.2 BASELINES

- **FERMI (Lowy et al., 2021):** Here we have used the official implementation of FERMI which can be found in the following link: `https://www.dropbox.com/scl/fo/tz8aksm4ibsta9l9hzig7/AMK3ixeUQRqoY0FhWgDy5rM?rlkey=yufnfhuvhs91mvvl9kc3lbss1&e=1&dl=0`. Here we have used the FERMI loss with the usual regularized cross-entropy loss to fine-tune the pre-trained model for E=1.
- **Continuous Fairness (Mary et al., 2019):** The official implementation can be found at: `https://github.com/criteo-research/continuous-fairness`. We

have used the usual regularized cross-entropy loss to fine-tune the pre-trained model for E=1.

- **Fariness-KDE (Cho et al., 2020a):** As there is no official implementation for this method we use the open-source implementation from `https://github.com/Gyeongjo/FairClassifier_using_KDE`. Similarly, like other baselines, we train the pre-trained model using this regularized loss for E=1.

