# OpenReview forum: "Concept forgetting via label annealing"
_ICLR.cc/2025/Conference — ICLR 2025 Conference Withdrawn Submission_

### Official Review · Reviewer_MEHY · 2024-10-16

**Soundness:** 1
**Presentation:** 3
**Contribution:** 2
**Rating:** 3
**Confidence:** 3

**Summary:**

This paper presents a new approach to studying concept forgetting, which aims to remove some concepts from pre-trained models while preserving their performance. To achieve this goal, the authors propose an algorithm called Label ANnealing (LAN), which employs a two-stage process to align the distribution of pseudo-labels with the class distribution, as generated by the trained model's predictions. Experimental evaluations on four benchmark datasets – MNIST, CIFAR-10, miniImageNet, and CelebA – demonstrate that concept violation can be effectively mitigated.

**Strengths:**

1. The idea of the paper is good and important for research.
2. The example in the introduction is also interesting that "envision a CelebA (Liu et al., 2015) image classifier that heavily relies on background color as a distinguishing feature to classify different celebrities, limiting its ability to generalize effectively".

**Weaknesses:**

1. Following the example provided in the introduction, I anticipated an improvement in performance after removing harmful features. Nevertheless, my findings contradict this expectation: despite claims of 'maintaining the model’s overall performance and generalization ability', I observed a significant drop in performance on all datasets, with a particularly notable 15% decrease on CelebA for the task 'Heavy makeup or not'. This discrepancy suggests that the authors should revisit their method to ensure it meets its stated objectives.
2. The concept of 'concept violation' is not rigorous, as it only evaluates model outputs without considering the nuanced effects of concepts within decision-making processes. Even when results appear identical, it is uncertain whether a particular concept has been entirely eliminated or merely masked in some way.
3. The alogrithm Label ANnealing is simple.

**Questions:**

1. I am not sure I fully understand the experiments. Are examples in forgetting classes removed, and examples in the rest of the classes are used to train and test?
2. I suppose the introduction example 'background' is good; I think in experiments, the authors should give the results of the example. Does the method only work with concepts that have labels? If so, this is a strong limitation to the proposed method.

---

> ### Author Response · Authors · 2024-11-23
> **Rebuttal by Authors**
>
> We are thankful for your positive comments, feedback, & suggestions. Specific doubts & questions are answered below.
> 1. **Accuracy improvement after forgetting:**
>     -  Achieving better performance after concept forgetting depends on the usefulness of the feature targeted for forgetting. Inclusion of any biased/harmful concept/feature might increase the classification accuracy. However, including such features are undesired. In our CelebA experiments, classifying images as young vs not-young facial hair is a useful feature because young people generally don't have facial hair. Thus forgetting the facial hair concept will reduce the classification accuracy. However, our goal is to show that LAN is effective in reducing concept violation of facial hair while maintaining a reasonable amount of accuracy intact.
>     -  This reduction of test accuracy is indeed expected as we are learning with further constraints of low concept violation. This type of phenomenon is also observed in fairness literature[1]. Further due to the effect of *catastrophic forgetting* [1,2,3] where adapting a model for new tasks can significantly degrade performance. In our case, the older task of retaining the pre-trained model's performance is traded off with the newer task of reducing concept violation.
> 2. **Concept violation is unaware of overall decision-making:**
>     -  Due to the hardness of mathematically defining a concept/feature's effects on model's overall decision-making process, the mathematical definition of concept violation is only restricted to the model's end predictions. Our motivation for this restriction is due to the fact that in many practical cases, the most important criterion is model's output predictions.
>    - Concepts are not masked at the output. With the goal of making concept violation zero, we fine-tune the initial model's parameters to generate the forgotten model effectively eliminating the concepts from the model.
> 3. **LAN is simple**
>     -  We thank the reviewer for pointing this out! In fact, we believe this is more of an advantage than weakness. Since the method is simple, it is easy to use.
> 4. **Explanation of Experiments:**
>     -  No we don't remove forget class during training and testing. The experiments are mainly divided into two categories: binary concept forgetting ($m=2$) & multi-level or non-binary concept forgetting ($m>2$). In case of binary concept forgetting (Ref. lines 381-385 and Table 1) e.g. MNIST digit classification problem, the objective is to forget a particular class digit concept e.g. class-3 data. Thus here $c=0$ represents concepts of non-digit-3 data and $c=1$ represents concepts of digit-3 data. Similarly, for multi-level concept forgetting (Ref. lines 433-436 and Table-2) while classifying CelebA images as young vs. not-young, we aim to forget subtle feature concepts such as hair color ($c=0$ represents black hair, $c=1$ represents red hair, $c=2$ represents blue hair, $c=3$ represents grey hair) from the pre-trained models. It can be seen from Table 1 and Table 2 that the LAN method is effective in reducing concept violation while maintaining a reasonable amount of model accuracy. Further, it achieves better concept violation vs. accuracy trade-off (Reference: Figure 3) than other baseline methods making it a better choice for concept forgetting.
> 5. **Experiments on the 'background' examples**
>     -  Thanks for your suggestion. Due to lack of dataset with labeled background features we could not include such examples in the experiments. However, we will further try to incorporate experimental results for forgetting 'background' concepts from a pre-trained model in the final version.
> 6. **Concepts with labels**
>     -  Even though current experiments show forgetting concepts that are labeled, our methodology is generic and applicable to concepts without labels. However in the case of concepts that are not labeled, one needs to manually identify the examples in the dataset where the concepts are present. For example, in the case of cat vs. dog classification if one wants to forget the 'background' concept ($c=0$ for indoor background and $c=1$ for outdoor background) where there is no indication of the background type for each example it becomes a harder task to find these two types of background concept. After identifying such concepts, LAN is effective in forgetting the concept.
>
>
> [1] Lowy et al. A stochastic optimization framework for fair risk minimization. arXiv, 2021
>
> [2] Goodfellow et al. An empirical investigation of catastrophic forgetting in gradient-based neural network. In Proc. of ICML, 2014
>
> [3] Kirkpatrick et al. Overcoming catastrophic forgetting in neural networks. Pre-print arXiv, 2017
>
> [4] Ginart et al. Making ai forget you: Data deletion in machine learning. In Proc. of NeurIPS, 2019.
>
> **If you are satisfied with our answers, please support our work by increasing the rating**

---

> > ### Comment · Reviewer_MEHY · 2024-11-25
> >
> > Thank you for the careful responses.
> >
> > 1. I acknowledge that incorporating useful features as concept forgetting can potentially impact test accuracy. Nevertheless, I believe it would be beneficial for the authors to provide explicit examples demonstrating how this proposed method can effectively eliminate or mitigate harmful features. For instance, in Review W77M, the removal of background features from "dogs are often photographed outdoors" should result in improved or comparable performance.
> >
> > 2. Since there is currently no evidence supporting the claim that the proposed technique maintains performance, I suggest a more rigorous examination of the concept of forgetting using visualization techniques. This could help us better understand why there is no improvement and identify potential issues with the method, such as negative effects during decision-making in the middle layers, even when the output is zero.

---

> ### Author Response · Authors · 2024-11-28
> **Further Clarification by Authors**
>
> **Maintain performance:** The goal is to reduce concept violation (ideally to zero) while maintaining the model's performance. Concept forgetting is a constrained learning problem where the goal of not only to maintain good performance but also to satisfy the constraint of low concept violation. Effectively one can image a restricted hypothesis space where the hypothesis must satisfy low concept violation. Now if you reduce the constraint i.e. concept violation increases the accuracy should increase because hypothesis space becomes large resulting us a better hypothesis with high accuracy. As we reduce the concept violation to zero this performance drop is expected.

---

> > ### Comment · Reviewer_MEHY · 2024-11-29
> >
> > Again, I wish the authors provided examples whose concept violation negatively affects the performance, and removing them will improve the performance. The authors should not focus on special causes where concept violation increases the accuracy. This may be a dataset problem.

---

### Official Review · Reviewer_W77M · 2024-10-30

**Soundness:** 2
**Presentation:** 1
**Contribution:** 2
**Rating:** 1
**Confidence:** 3

**Summary:**

The author proposes a new issue termed concept forgetting.
The author argues that, to forget a concept, the label proportions should be constant regardless of the concept.
The author proposes an approach in which, when the label distribution varies according to a specific attribute in a pre-trained model, this is directly adjusted before further training.

**Strengths:**

The author has proposed an intriguing problem.
If concept forgetting is feasible, it may also be possible to remove unwanted information from a pre-trained model.

**Weaknesses:**

First, the proposed problem appears to be an ill-posed problem.
According to the author’s assertion, the entire dataset must be pristine.
If there is a concept not included in the dataset or if certain concepts are overrepresented, the optimal model for concept forgetting will be defined differently.
In fact, consider the example commonly addressed in debiased classification: in the dog and cat problem, dogs are often photographed outdoors, while cats are typically photographed indoors.
If additional outdoor photos are included, the label for indoor cats would need to be even more frequently replaced with that of dogs in the author’s algorithm.

Secondly, despite the author’s algorithm being highly intuitive and straightforward, its characteristics are not well explained.
The author replaces explanations of the proposed method with figures and algorithms, which does not aid intuitive understanding.
Even concept forgetting is not well explained beyond the measure defined as concept violation.
At the very least, it would be essential to verify whether the author’s method is beneficial when solving zero-shot classification tasks that align concepts in the trained model.

Lastly, in the theoretical analysis, the gap between the two terms in the inequality is substantial.
For the theoretical analysis to be meaningful, this gap needs to be minimized; the current gap arises from using the maximum value of the loss.
In the case of cross-entropy loss, the bound is exceedingly large, and when multiplied by the concept violation values observed by the author in Table 1, the upper bound of the curated loss inevitably becomes significantly large.
In fact, it is challenging to identify a clear correlation between the concept violation values and the reduced accuracy in the experimental results.

**Questions:**

(Clear problem definition)
Can the author explain the purpose of the algorithm with a real-world example? I did not intuitively grasp the goal of concept forgetting. For instance, I am curious about a plausible purpose, such as removing privacy-sensitive information.
Furthermore, the issue I mentioned in the weaknesses section, where the optimal solution for concept forgetting changes if the entire dataset changes, indicates that concepts may not be fully removed when a larger, pristine global dataset exists beyond the given dataset. I am curious about the author's assumptions regarding the entire dataset in this context.

(Justification of the measure)
Additionally, while concept violation appears to be a reasonable measure, it does not necessarily reflect whether concept forgetting has truly been achieved. Cross-entropy loss is a good measure for classification tasks, but for models trained with techniques like label smoothing, the loss can increase independently of accuracy. Similarly, I believe that concept violation cannot be considered a perfect measure. Since concept violation is a measure introduced by the author, it requires thorough analysis from multiple perspectives; however, in the submitted paper, it is only used as a measure without further analysis. It seems necessary to include a qualitative analysis in the experiments demonstrating that low concept violation indeed addresses the intended purpose of concept forgetting. In addition to the analysis I suggested, any results that can further demonstrate the utility and significance of your concept violation measure would be welcome.

(Representation)
The methods for the author’s algorithm can all be represented by figures and pseudo code. This implies that Section 4.1 is somewhat redundant. Adding insights into each step of the algorithm in the main text would be beneficial. For example, is the sorting in line 4 truly meaningful? What is the reason for selecting the next label deterministically in line 9? What is an adequate range for E? Addressing questions like these would enable a deeper understanding of the author’s algorithm.
Lastly, the author’s theoretical analysis does not provide much help in interpreting the experimental results. Is it possible to define a tighter boundary under specific conditions?

---

> ### Author Response · Authors · 2024-11-23
> **Rebuttal by Authors**
>
> 1. **Ill-posed Problem**
>     -  We strongly refute this claim. The problem is mathematically well-defined. We propose a clear definition of concept forgetting by defining *concept neutral* & propose  *concept violation* metric that measures the extent of concept forgetting. Now, the goal is to reduce the concept violation while retaining the model's accuracy.
> 2. **Pristine Dataset & Assumption on dataset**
>    - This point is unclear - please further clarify. We don't assume anything about the dataset. The concept of forgetting is defined by the model's prediction on the dataset. All we assume is that we have a pre-trained model with good accuracy.
> 3.  **Cat vs. Dog classification**
>     - This point is also unclear - please further clarify. Here the task will be to classify images as dog vs. cat while forgetting the 'background' concept. Thus $c=0$ denotes indoor background & $c=1$ denotes outdoor background. Now LAN algorithm will try to reduce empirical concept violation to zero in $\mathcal{D}_c$ for each $c \in (0,1)$ and then fine-tune the pre-trained model to produce the forgotten model.
> 4. **Characteristics of LAN**
>     -  Apart from the LAN algorithm's intuitive explanation using the figure & detailed algorithmic insights for the algorithm (Ref. Sec 4.1), detailed theoretical characteristics of the label annealing subroutine in Algorithm 1 are well analyzed in Lemma-1 (please look into Appendix section A). From this lemma, it can be seen that the total no of labels changed by the LAN algorithm is in the order of $O(n \cdot \hat{V}(\theta^*,\mathcal{C},{\mathcal{D}}))$. Now for the second stage (Algorithm 2), we provide a theoretical upper bound on the accuracy of the forgotten model.
> 5.  **Concept forgetting is not well explained and Analysis of concept violation**
>     -  Concept forgetting is a broader term and one of the goals of our work is to mathematically define concept forgetting. Our definition of forgetting is motivated by the fact that in the case of humans if one forgets a concept, the forgotten concept doesn't affect one's decision-making. Thus concept forgetting within the context of machine learning entails ensuring that a model's predictions become entirely independent of the targeted forgetting concept. With this motivation, we propose the definitions of *concept neutral* & *concept violation* to characterize concept forgetting mathematically.
>     -   Please note that the goal of *concept forgetting* is to achieve zero concept violation while retaining a reasonable amount of accuracy. Experimental evidence suggests that LAN achieves these objectives. Please further clarify what sort of qualitative analysis is required.
> 6. **Zero-shot classification tasks**
>      - LAN assumes access to the original dataset to produce the forgotten model and thus needs samples from the original dataset (not a zero-shot case).
> 7. **Theoretical Gap:**
>     -  The upper bound on Theorem-1 can be large because of using the maximum value of the loss. This bound is represented in generic terms i.e. without any further assumption on the functional form of loss. However, we refute the claim that this upper bound is very loose. In particular cases, this bound can be well attained e.g. if the original concept violation is close to zero, then labels changed by the label annealing sub-routine in Algorithm-1 is zero, making the loss of the forgotten model (in expectation over SGD steps of algorithm 2) is the same as the loss of the original model.
> 8. **Correlation between concept violation and accuracy:**
>    - There is a clear positive correlation between concept violation and accuracy. As accuracy increases so does the concept violation marking a clear trade-off between these two metrics (Ref. lines 410-412 and Figure 3).
> 9. **Real-world Usecases**
>     -  To ensure fairness and accountability, it is crucial to forget these biased/undesired concepts from trained models. Further, concept forgetting can enhance domain generalization. For instance, a CelebA classifier might over-rely on background color, hindering its ability to generalize. Thus forgetting background concepts is useful.
> 10. **Goal of concept forgetting:**
>     -  The goal of concept forgetting is to reduce concept violation (ideally zero) while retaining model's initial performance. Due to the hardness of removing features from a model we try to mathematically define concept forgetting and propose concept violation as a metric for concept forgetting. Also, concept forgetting should be computationally inexpensive (Ref. lines 219-232).
> 11. **Intuitive Understanding & Insights on the algorithm:**
>     -  Thanks for acknowledging that LAN is highly intuitive and straightforward. The figures and algorithms aid this intuitive understanding.
>     -  We have modified the explanation of LAN algorithm in section 4.1 with insights for each steps. Please take a look.
>
> **If you are satisfied with our answers, please support our work by increasing the rating.**

---

> > ### Comment · Reviewer_W77M · 2024-11-25
> >
> > I sincerely appreciate the authors' comprehensive and well-articulated rebuttal aimed at addressing the concerns I raised.
> > The authors have proposed an intriguing objective, and the manuscript provides supporting arguments for it.
> > However, my concern is centered on whether the proposed objective holds significant practical or theoretical value.
> > If the authors can convincingly address this concern, I would regard this paper as a valuable contribution to the community, presenting a novel problem in the context of concept forgetting.
> > I have outlined below the remaining concerns.

---

> ### Comment · Reviewer_W77M · 2024-11-26
>
> 1) The Meaning of the Objective
>
> The authors claim that the problem they define is mathematically well-defined through the proposed objective. However, I find this assertion difficult to accept.
>
> The authors' motivation is clearly articulated in line 62: to maintain performance while forgetting a concept. Are the authors confident that minimizing the proposed objective is equivalent to achieving this goal?
> In fact, if the model's output remains constant regardless of the input, the concept violation is minimized. Thus, the objective appears to be merely a metric for quantifying the extent of concept forgetting, rather than a rigorous mathematical model of the authors' motivation.
>
> If there is a misunderstanding on my part regarding the proposed objective, I would appreciate further clarification. Additionally, I request a detailed explanation of the correlation between minimizing the objective and maintaining model performance.
>
> 2) Limitations of Concept Neutrality
>
> As I outlined in my first review, the proposed objective is defined as a loss function that enforces the prediction distribution of the global dataset to be similar to that of the local (concept-specific) datasets, making it inherently dependent on the overall dataset.
> To strengthen the robustness of the proposed problem, the authors need to establish mathematical (or at least semantic) assumptions about the overall dataset.
>
> For example, in the dog-and-cat problem I mentioned, the dog and outdoor concepts are strongly correlated. In such cases, if the overall dataset contains significantly more dog samples than cat samples, the proposed method may relabel most of the cat samples in the indoor concept as dogs. Ultimately, the model might be trained to output only "dog." Contrary to the authors' claim that Theorem 1 is meaningful, in this scenario, even if concept violation converges to 0, the accuracy will inevitably drop significantly. The authors should recognize that cross-entropy has no upper bound.
> This scenario is likely not what the authors intended.
>
> I recommend that the authors introduce minimal assumptions about the overall dataset to prevent such situations. For instance, in the proof of the theorem, the authors leverage the bounds of the loss function, and these bounds could be tightened through mathematical assumptions about the overall dataset.
> As I mentioned in my first review, the current gap is substantial. To reiterate, in a scenario where the loss has no upper bound, the convergence of concept violation to 0 is not meaningful.
> Clearly defining the conditions the authors consider would likely address this concern.
>
> 3) Representation
>
> I find the authors' rebuttal on this issue unconvincing. At a minimum, the authors should provide insights into line 4, line 9, and E (more informative than line 339) as mentioned in my first review. Without this, I cannot trust that the authors will adequately address the representation issue.
>
> 4) Other Limitations
>
> I remain unconvinced by the rebuttals addressing concerns about the motivation and practical applicability of the proposed method, which other reviewers have also flagged. I will refer to the authors' responses to the other reviewers' concerns when determining my final evaluation score.

---

> ### Author Response · Authors · 2024-11-28
> **Further Clarification by Authors**
>
> Thanks for your suggestions.
>
> 1. **Meaning of the objective:**
> - This work has a clear goal: *forget concepts that are undesired without hurting the model's performance*. Concept violation which is the metric for concept forgetting is mathematically well defined. Thus the goal is to **reduce concept violation (ideally to zero) while maintaining the model's performance.**
> - Yes we are confident that reducing concept violation while maintaining model performance is the true goal for concept forgetting. Please note that as we previously stated, we assume that we have a pre-trained model which has high accuracy. Now to answer your question, if the model's output remains constant regardless of the input the pre-trained model has very low accuracy. However in this case model gives output irrespective of input meaning it is already independent of any concept/feature present in the input. Thus no need to forget the concept in the first place. Please also note that these kinds of scenarios do not generally happen. Assuming a pre-trained model that has some dependence on input features is common and that is the primary motivation for the definition of concept neutral and concept violation.
> - *Correlation between concept violation and maintaining model performance:*  concept forgetting is a constrained learning problem where the goal of not only to maintain good performance but also to satisfy the constraint of low concept violation. Effectively one can image a restricted hypothesis space where the hypothesis must satisfy low concept violation. Now if you reduce the constraint i.e. concept violation increases the accuracy should increase because hypothesis space becomes large resulting us a better hypothesis with high accuracy. Thus there's a clear positive correlation which is also visible in the experiments.
>
> 2. **Assumption of Dataset:**
> -  The definition of concept violation depends on the prediction of the model on the global dataset. Nothing further is assumed about the global dataset.
>
> 3. **Representation:**
>  - As stated in the rebuttal we have already incorporated these changes in Section 4.1. The insights of line-4 are given in lines 312-313. The insights of line line-9 are given in lines 313-314. The insights for the values of E are given in lines 321-323.
>
> 4. **Motivation and Practical applicability:**
> -  Our motivation for the proposed method lies in making model predictions independent of the concepts. In this case, we assume when a model forgets a concept its prediction is independent of the concept.
> -  LAN can be applied to a wide range of concept-forgetting cases where we have access to the original dataset.

---

> > ### Comment · Reviewer_W77M · 2024-11-28
> >
> > Unfortunately, despite the authors' kind responses, most of my concerns remain unresolved. Therefore, I will maintain my initial rating.

---

### Official Review · Reviewer_aSbg · 2024-11-01

**Soundness:** 2
**Presentation:** 2
**Contribution:** 2
**Rating:** 3
**Confidence:** 4

**Summary:**

The paper proposes a novel approach for concept forgetting in deep neural networks. For this purpose, they introduce a two-stage iterative algorithm called Label Annealing (LAN). In the first stage, pseudo-labels are assigned to the samples by annealing or redistributing the original labels based on the current iteration’s model predictions. In the second stage, the model is fine-tuned on the dataset with pseudo-labels. They also introduce a novel metric called 'concept violation' that measures how much the model forgets a specific concept. The proposed algorithm has been validated across various models and datasets.

**Strengths:**

- The paper addresses a very relevant topic nowadays related with data privacy, which is represented by machine unlearning
- The paper presents a novel approach for concept forgetting in deep neural networks
- The related work covers most of the relevant paper in the field

**Weaknesses:**

- the paper is difficult to read, the clarity of both text and figures should be significantly improved
- the experimental validation is limited and not convincing. The authors compare their approach against 3 baselines, and none of them is related with concept forgetting

**Questions:**

Here are my concerns:
-  The differences between concept forgetting and machine unlearning are mentioned at the end of section 2. The authors should clarify this differences much earlier, in the introduction.
- Regarding definition 1:  'c' represents a class label or a feature?
- Regarding LAN algorithm: Why do you need to assign pseudo-labels? How do you deal with errors in pseudo-label assignment? Why don't just remove the classifier head corresponding to the removed concept?
- The problem of concept forgetting relies not only in retraining the classifier. The knowledge associated with it is implicitly embedded into the network's weights. How do you remove the information related with the concept being forgotten from the network's weights? I have not seen any discussion about this. If you retrain the network with the remaining data (after extracting the concept to forget), then this solution is trivial. What if the original data (used to train initially the network) is no longer available?
- Section 5.5: What means multi-level concept forgetting? Do you assume data is multi-labeled?
- In the experimental results, compare your approach against some methods from the current state of the art.

---

> ### Author Response · Authors · 2024-11-23
> **Rebuttal by Authors**
>
> We are thankful for your positive comments, feedback, & suggestions. Specific doubts & questions are answered below.
> 1. **Clarity on text & figures**
>     -  Apologies. We will improve the figures in the final version. For the text part, we have modified explanation of LAN algorithm in section 4.1. Please take a look and let us know if some other parts need modification.
> 2. **Limited experiments & State-of-the-art Baselines**
>     -  We are open to incorporating further experimental suggestions. In current experiments, we cover both binary concept forgetting ($m=2$) & multi-level or non-binary concept forgetting ($m>2$). We have experimented with different classification models such as 2-layer-MLP, Mobinetv2, Densenet-121, Resnet-50 on both lower-dimensional datasets such as MNIST, CIFAR-10 & high-dimensional datasets such as miniImageNet (image dim: $224\times224$), CelebA (image dim: $178\times218$).
>     -  We are open to incorporating further baseline suggestions. According to our knowledge, this is the first work that introduces *concept forgetting* property that induces independence from the forgetting feature during its prediction task. Thus we are unaware of any such state-of-the-art baselines for concept forgetting.  However, we adapted three state-of-the-art baselines from fairness because these baseline methods also advocate for the independence of prediction and unfair concept features.
> 3. **Difference between concept forgetting & machine unlearning:**
>     -  Please note that we have already explained the difference between concept forgetting and machine unlearning in great detail in the Introduction section 1 (Ref. lines 68-95).
> 4. **Meaning of c**
>     -  Here $c$ is a particular concept/feature value to be forgotten. The concept targeted for forgetting defined as $\mathcal{C}$ takes multiple values $c \in (0,1,...,m-1)$. Now, for binary concepts ($m=2$) such as *beard*, $c=0$ denotes the absence and $c=1$ denotes the presence of the beard. For non-binary concepts such as *facial hair*, $c \in (0,1,2,3)$ ($m=4$) signifies no facial hair, mustache, beard, and goatee respectively.
> 5. **Need for Pseudo-Labels and Error in Pseudo Labeling:**
>     - Pseudo labels are useful to create a dataset $\widetilde{D}$ where the empirical concept violation is zero. In order to reduce the concept violation, the pre-trained model is fine-tuned on this pseudo-labeled dataset $\widetilde{D}$.
>     - There is no error in pseudo label assignment because the label annealing subroutine in Algorithm 1 is deterministic in the sense that it assigns pseudo labels based upon the current model's prediction on a particular iteration.
> 6. **Removal of classifier head:**
>     -  Removing a classifier head doesn't signify the model has forgotten a concept. Further, it is not always possible to remove the classifier head corresponding to the forgetting concept. Suppose in cat vs. dog classification, one wants to forget the concept of background (binary concept with c=0 signifies indoor background & c=1 signifies outdoor background). Now, classifying images as cat vs. dog, there is no classifier head corresponding to the background. Thus making this solution ineffective.
> 7. **Evaluation in weight space**
>     -  Due to the high dimensionality of weight space and unexplainable correspondence between input features and the weight, it is hard to evaluate if the forgotten feature is effectively been removed from the weights or not. Thus a modest goal of concept forgetting is to remove the dependence on the forgetting feature from the model's prediction in output space. This dependence is quantified by empirical concept violation and achieving zero concept violation is indicative of achieving concept forgetting.
> 8. **Retraining**
>     -  To forget a concept with the method of retraining is limited and not always feasible. In the case of tabular datasets removing the undesired features from the data and retraining the model seems trivial. However, this trivial method is hard and sometimes infeasible in the case of image and text datasets. Due to the high entanglement of the different concepts/features, it is not always possible to extract and remove the undesired features from the dataset. Thus retraining seems infeasible. Therefore this work tries to propose an algorithm that tries to forget concepts with a modest goal of removing the dependence of forgetting features from model's prediction.
> 9. **Unavailability of original data**
>     -  In our case, we assume availability of original data.
> 10. **Multi-level concept:**
>     -  In multi-level concept forgetting the forgetting concept is non-binary i.e. $\mathcal{C}(z)=c \in (0,1,...,m-1)$ with $m>2$ (Ref. lines 418-419).  For example, if the forgetting concept is *facial hair* then it can take multiple values $(0,1,2,3)$ ($m=4$) which signifies no facial hair, mustache, beard, and goatee respectively.
>
>
> **If you are satisfied with our answers please support our work by increasing the rating**

---

### Official Review · Reviewer_4rao · 2024-11-04

**Soundness:** 2
**Presentation:** 3
**Contribution:** 2
**Rating:** 5
**Confidence:** 5

**Summary:**

To enhance the safety and responsibility of machine learning, this paper introduces a new task, concept forgetting. To achieve the goal of forgetting specific concepts while retaining the general ability of the original model, authors develop an iterative two-stage algorithm. The core idea of the algorithm is to ensure zero concept-violation on the newly created dataset by redistribution and relabeling.

**Strengths:**

- This paper proposes a novel and interesting problem referred to as concept forgetting. The task is set to forget a specific undesired concept without degrading the general ability. It is similar to the opposite counterpart of catastrophic forgetting but has not been well studied.
- The coherent text and the smooth transitions strengthened the readability of this paper.

**Weaknesses:**

- It’s difficult to understand the explanation of Algorithm 1, eg. in line 311 to line 315.
- As shown in Table 1, there is still an obvious reduction in test accuracy. I recommend more analysis of the reasons.

**Questions:**

- I have doubts about whether the algorithm has achieved a good experience effect. Firstly, it is because of the lack of enough competitors. Secondly, it is about the trade-off between concept violation and accuracy: if a concept is forgotten, the network should theoretically achieve better performance on other concepts.
- Have you considered the trade-offs between increasing the number of iterations (E) and maintaining model accuracy?

---

> ### Author Response · Authors · 2024-11-23
> **Rebuttal by Authors**
>
> We are thankful for your positive comments, feedback, and suggestions. Specific doubts and questions are answered below.
>
> 1. **Explanation of Algorithm-1:**
>     - Apologies. We have further clarified the explanation of the algorithm in lines 311-315 (section 4.1). Please take a look.
>
> 2. **Reduction in Test accuracy:**
>     - This reduction of test accuracy is indeed expected as we are learning with further constraints of low concept violation. This type of phenomenon is also observed in fairness literature[1] where the incorporation of additional constraints reduces the model's performance.
>    - Further, this phenomenon of lower accuracy can be explained by the effect of *catastrophic forgetting* [2,3,4] where adapting a model for new tasks can significantly degrade performance. In our case, the older task of retaining the pre-trained model's performance is traded off with the newer task of reducing concept violation.  (Reference: lines 54-59).
>
> 3. **Lack of enough competitors:**
>     -  We are open to incorporating further baselines. Please let us know. According to our knowledge, this is the first work that introduces *concept forgetting* as a property of the forgotten model to induce independence from the forgetting feature during its prediction task. Thus we are unaware of such baseline methods for concept forgetting. However, for comparative evaluation, we adopt three state-of-the-art baselines from fairness because these baseline methods also advocate for the independence of prediction and unfair concept features. Thus, these baselines are included (Reference: lines 370-377).
>
>  4. **Better performance after concept forgetting**
>     -  Achieving better performance after concept forgetting depends on the usefulness of the feature targeted for forgetting. Any biased/harmful feature can be useful for certain prediction tasks i.e. inclusion of such a feature might increase the classification accuracy. However, including such features are undesired. For example, suppose we are learning a model to predict whether a person should get a bank loan or not. Such a model should not depend on the gender or race of the person. However, it is possible that the machine learning model might inadvertently use these features to make better predictions (Reference: lines 39-42). Similarly, in our experimental settings (Reference: Table 2), classifying images as young vs not-young facial hair is a useful feature because young people generally don't have facial hair. Thus forgetting the facial hair concept will reduce the classification accuracy. However, our goal is to reduce the dependence (concept violation) on the facial hair concept while maintaining a reasonable amount of accuracy intact.
>
> 5. **Trade-off at higher values of E:**
>     -  Thanks for this suggestion. In Figure 4 (Reference: Section 5.6 Ablation studies lines 485-504), we demonstrate the effectiveness of the *LAN* algorithm over multiple iterations (E=2, E=4). As E increases at higher accuracy regions, the concept violation further decreases for the same accuracy value making the trade-off plot flatter. This indicates as we increase the values of E, a better trade-off between concept violation and model accuracy is achieved.
>
> [1] Lowy et al. A stochastic optimization framework for fair risk minimization. arXiv, 2021
>
> [2] Goodfellow et al. An empirical investigation of catastrophic forgetting in gradient-based neural network. In Proc. of International Conference on Learning Representations, 2014
>
> [3] Kirkpatrick et al. Overcoming catastrophic forgetting in neural networks. Pre-print arXiv, 2017
>
> [4] Ginart et al. Making ai forget you: Data deletion in machine learning. In Proc. of NeurIPS, 2019.

---

> ### Comment · Reviewer_4rao · 2024-11-26
>
> Thank you for your detailed response. The authors have revised the presentation regarding Algorithm 1 and considered the trade-offs at higher values. However, three key concerns remain unaddressed:
> 1. Limited Motivation. Concept forgetting is proposed as an interesting technique to improve performance by removing undesired concepts. However, the experimental results show consistent degradation in performance across all settings. Since the authors consider this outcome to be expected, it is difficult to support the strarting point of this method.
> 2. Practicality of the Algorithm. LAN attempts to recreate a dataset through multiple iterations of assigning pseudo-labels, which is computationally expensive. Why not simply select samples that are unrelated to the forgotten concept, thus reducing the need for such a costly procedure?
> 3. Unconvincing Experimental Results. The paper's goal is ambiguous. If the aim is not just to reduce empirical concept violations, but to propose a new solution in an area with limited competitiors, then it is crucial to demonstrate the method’s effectiveness in a practical or realistic scenario.
>
> Given these points, I am lowering my initial score.

---

> > ### Author Response · Authors · 2024-11-28
> > **Further Clarification by Authors**
> >
> > 1. The motivation of this work is clear: forget concepts that are undesired without hurting the model's performance. Please note that the concept forgetting is now a constrained learning problem where the constraint is to reduce the concept violation. The addition of this low-concept violation constraint reduces model accuracy (effectively reducing the hypothesis space where concept violation is low). The starting point of concept forgetting was to reduce concept forgetting without much hurting the model performance. Experimental evidence suggests that LAN is effective in achieving this goal.
> >
> > 2. Selecting samples unrelated to the forgetting concept is not feasible. In case of gender concept forgetting, if the dataset contains male and female images all samples have gender concepts. LAN is computationally inexpensive because of its effectivity in one-iteration (E=1) only.
> >
> > 3. Further experimental suggestions are welcome. The goal of this work is to reduce concept violation without much hurting model's performance.  Current experiments are realistic e.g. forget gender concept while determining attractiveness or forgetting hair color in determining young vs not. young.  The current comparison is done on three state-of-the-art baselines. The goal of current experiments is to establish the effectiveness of LAN in various concept-forgetting scenarios.

---

### Note · Authors · 2025-01-23

I have read and agree with the venue's withdrawal policy on behalf of myself and my co-authors.